# Effectiveness of Physical-Literacy-Based Online Education on Indices of Physical Fitness in High-School Adolescents: Intervention Study during the COVID-19 Pandemic Period

**DOI:** 10.3390/children10101666

**Published:** 2023-10-08

**Authors:** Barbara Gilic, Mirela Sunda, Sime Versic, Toni Modric, Dragana Olujic, Damir Sekulic

**Affiliations:** 1Faculty of Kinesiology, University of Split, 21000 Split, Croatia; barbara.gilic@kifst.eu (B.G.); sime.versic@kifst.eu (S.V.); toni.modric@kifst.eu (T.M.); dragana.olujic@nutricionist.hr (D.O.); 2Faculty of Kinesiology, University of Zagreb, 10000 Zagreb, Croatia; mirela.sunda@kif.hr

**Keywords:** physical literacy, physical activity, nutrition, lifestyle medicine

## Abstract

There is a global consensus that physical literacy (PL) is an important determinant of physical fitness (PF), but studies have rarely examined the effects of PL-based interventions on PF. The aim of this research was to evaluate the effectiveness of specific online video-based PL intervention on PF indices in high-school students from Croatia. Participants were 423 high-school adolescents (295 girls, 128 boys; 14-to-18 years of age), divided into an intervention group (*n* = 230) and a control group (*n* = 193). The intervention lasted 12 weeks. Educational video materials were disseminated to the intervention group by the closed social network during the pandemic period. Variables included height, mass, BMI, cardiorespiratory fitness (CRF), and power, strength, and flexibility indices. Pre- to post-testing design was applied, with two-way analysis of variance for repeated measurement (Time × Group). Applied intervention induced positive effects in CRF (the intervention group improved their capacities, while no changes occurred in the control group) and BMI (the intervention group retained their BMI levels at the pre-testing level, while BMI of the control group slightly increased over the course of the study), with better effectiveness in girls than in boys. No effects were evidenced for other variables. The positive effects of applied educational intervention on BMI and CRF are encouraging knowing that both indices are related to health status.

## 1. Introduction

Adequate physical activity (PA) is one of the essential determinants of health status and is a pillar of lifestyle medicine [1,2,3]. However, there is a great prevalence of people with insufficient physical activity levels (PALs), with the most alarming situation among adolescents (i.e., 81% of adolescents globally do not have sufficient PALs) [4]. Namely, the biggest drop in PAL happens during the transition from childhood to adolescence; it appears at around 14 years of age and linearly drops during adolescence [5].

Another significant indicator of health status is physical fitness (PF). PF is a set of attributes which individuals have or attain that provides the ability to perform PA and is considered a measure of integrated body functions, including the cardiorespiratory, musculoskeletal, and metabolic systems [6]. It is worth noting that PF and its facets (i.e., musculoskeletal fitness, cardiorespiratory fitness (CRF)) have several positive health outcomes in children and adolescents. Moreover, PF during adolescence was associated with a healthy cardiovascular disease risk profile in adulthood, which emphasizes the importance of developing healthy and optimal PF during adolescence [7].

However, despite the well-known positive effects of PA on PF and health in general, the issue of inadequate PA is still eminent. Thus, it was proposed that adolescents lack motor proficiency and general knowledge of correct PA patterns and that other constructs should be developed to target an increase in PAL. Such a “background” concept is widely named physical literacy (PL). PL is a multidimensional concept defined as a “disposition to capitalize on our human embodied capability, wherein the individual has the motivation, confidence, physical competence, knowledge and understanding to value and take responsibility for maintaining purposeful physical pursuits/activities throughout the life course” [8]. PL consists of four domains, including physical, cognitive (knowledge and understanding), affective (confidence and motivation), and behavioral domains [9]. Several studies noted a correlation between PL, PF, and PAL in youth, with increased PL leading to better PF and higher PAL [10,11].

A recent review study collected all evidence on the effectiveness of PL interventions, including the ones conducted among adolescents [12]. Specifically, a study on Canadian adolescents aged 17.85 ± 0.51 years that evaluated the effects of a 12-week design to facilitate new movement skills recorded an enhancement of overall PL, including motivation knowledge and understanding subdomains [13]. Moreover, the PL intervention led to improvements in PA behaviors and CRF of first-year university students from Canada [14]. Meanwhile, there is an evident lack of studies that examine the effects of PL interventions on PF in adolescents.

Even though it is advisable to assess and make PL interventions targeting all domains of PL, in certain occasions, this is not possible. Specifically, during the COVID-19 pandemic and strict social-distancing measures, adolescents were attending online classes, or they had live classes but with social-distancing measures [15,16]. Thus, the opportunities for regular PL interventions and assessments were reduced. Hence, conducting PL interventions on forms of education materials (i.e., video materials, banners) targeting the knowledge and understanding domain of PL was considered the most appropriate in the given situation.

There is a global consensus that PL is an important determinant of PF, especially in childhood and adolescence. However, studies rarely examine the effects of PL-based interventions on PF, while, as far as we are aware, no study has examined similar issues among adolescents from southeastern Europe. The aim of this study was to evaluate the effectiveness of specific PL intervention (e.g., video-based-education) on PF indices (CRF, flexibility, muscular strength and power) in high-school students during the pandemic period. Knowing the specifics and differences of PF between genders, in this study, we applied a gender-stratified approach to study the effects of PL intervention on PF in high-school adolescents. We hypothesized that applied intervention would have positive effects on PF irrespective of gender. This research would give insight into how PL intervention impacts PF, including CRF, flexibility, and muscular strength and power, which are important determinants of health status in general.

## 2. Materials and Methods

### 2.1. Participants and Study Design

In this intervention study, 423 high-school adolescents (295 girls, 128 boys; 14-to-18 years of age) from Osijek-Baranja County in northern Croatia participated in both initial and final testing points. At the initial point of the assessment, 523 participants (366 girls, 153 boys) were included. All students were enrolled in the regular Croatian school system. The students were divided into an intervention group (*n* = 230) and a control group (*n* = 193), matched by gender using random class sampling. The division in groups was based on a random sampling of classes. Based on school records at the beginning of the 2021/22 school year, there was no significant group difference in out-of-school sport participation and students’ participation/non-participation in physical education (PE).

Participants from the intervention group were included in an educational intervention through a specially designed distance-learning program once a week for twelve weeks, in addition to the regular program of physical education (PE) classes, while respondents from the control group followed the regular PE teaching program and did not receive any intervention during research. The study was approved by the Ethical Board of the Kinesiology University of Zagreb, Croatia (Ref. no. 25/2021, date of approval: 16 July 2021). Parents or legal guardians signed informed consent for all participants who were under 18. Students with medical conditions were excluded from the investigation. The study design, including group assignments and interventions, is detailed in Figure 1.

### 2.2. Intervention Program

The intervention program in this study consisted of educational video materials and was developed by authors of the study, who are all long-term professionals and academicians in the field. Specifically, PE teachers and strength-and-conditioning coaches developed education materials for general PL issues (i.e., first two videos) and other videos covering various aspects of PF and specific fitness capacities (i.e., CRF, strength, flexibility)), while a nutritionist developed videos covering nutrition and diet (Appendix A). Educational videos were sent to the intervention group by a closed social network, specifically the Microsoft Teams platform, which was implemented by the Croatian Ministry of Education and Science as an official online communication and education platform during the COVID-19 pandemic period. All students participated in regular physical education, but only students who were individually logged into the Microsoft Teams group, where education materials were posted, participated in the intervention. However, following the experiment, all educational materials were made freely available on the open streaming channel. For the purpose of this article, educational materials are given as Appendix A with subtitles in the English language (note that original educational materials were prepared in the Croatian language).

The intervention was applied from October to December 2021. It included the application of specific and targeted short-term educational video materials (i.e., tutorials, lessons) aimed at improving the knowledge and understanding subdomain of PL. The intervention included information of the health benefits of PA, the importance of PA in general, the importance of proper nutrition, the development of cardiovascular endurance, methods of training of muscle power and strength, flexibility, and the possibilities to increase PA in limited spaces and with limited resources (i.e., in circumstances similar to the situation of the COVID-19 pandemic). Every week, new video material was posted on the Teams platform, but students were able to re-watch the videos previously posted.

The first video material that was disseminated in the first week of the intervention was about PL in general. In this video, the concept of PL was presented and defined as the importance of the domain knowledge and understanding of PL. In the second and third videos, CRF was covered. Students received information about when, how, why, and what to do to improve their CRF. Furthermore, they received information about the benefits of improving CRF for everyday life and the proposal of specific activities for its improvement. In the fourth video, the information from the second and the third video was repeated, but they were said and shown in a different way. In the fifth video, the muscle strength issue was presented. This video covered the aspects of the definition and importance of muscle strength, how to overcome resistance, and what strength depends on. In the sixth video, the benefits of strength training for the human body and health, as well as methods of strength training, were covered. In the seventh video, information from the fifth and sixth videos were repeated, but in a different way. In the eighth video lesson, the students received information about what flexibility is and how to improve it. The ninth video included information about the benefits of flexibility, what are the disadvantages of reduced flexibility, and flexibility training methods. In the tenth and eleventh video, materials about diet and nutrition were presented, consisting of information about types of food, what are real food products and how to prepare them, what is best to eat, what is obesity and how we see it, and how to protect themself from obesity. The topic of the last video material was PL again, in which we summarized all aspects and the importance of PL once again.

### 2.3. Variables and Measurements

Variables in this study were as follows: gender (male vs. female), study group (intervention vs. control) anthropometrics and indices of PF. All measurements and assessments were conducted in the school facilities (i.e., PE gym) during the morning (8:00–10:00 a.m.) to avoid diurnal variations. Anthropometric variables included body mass, body height, and body mass index (BMI), measured by standard procedures and equipment [17].

The PF variables included variables of muscular strength (sit-ups), flexibility (sit-and-reach test), power capacity (broad jump), and CRF (pacer test). All tests are regularly used in the Croatian school system, and their reliability has been proven previously [18].

The standing long jump test was used to assess jumping power capacity. The test was performed on a special surface for measuring the long jump with a centimeter scale (Ghia Sport, Pazin, Croatia). The subjects stood barefoot behind the starting mark in a hip-width stance and performed a maximal jump forward. The achievement was measured from the zero value to the footprint on the mat. The result was read in centimeters.

The purpose of the sit-and-reach test is to assess the flexibility of the lower back and hamstrings region, which is defined as the ability to perform the maximum amplitude of one movement. The test was performed using a wooden bench. The students sat on a surface without shoes, legs fully extended and touching the bench with the entire surface of their feet. They performed a maximum forward bend with a slow descent. The length of the maximum reach was measured on a centimeter tape. The result was read in centimeters [19].

The purpose of the maximum number of sit-ups in 60 s was to assess abdominal strength. The test was performed with the subjects lying down on the mat on their backs, knees bent, upper legs and lower legs at a 90°, and feet placed on the floor, and forearms and palms placed on the upper leg. At the signal, the students performed a sit-up and returned to the starting position. The result was recorded as the maximum number of correctly performed sit-ups in 60 s.

The purpose of the pacer test was to assess CRF. Students were in front of the line in a high start position facing the other line located 15 m away. The aim was to run a distance of 15 m within the sound signal, which determines running speed. The result is expressed numerically, depending on the number of levels and intervals run. After each beep, the audiotape pronounces the current level and interval. This is a version of the usual shuttle run test of 20 m because the shorter distance was considered more acceptable for children and adolescents [20].

### 2.4. Statistics

The normality of the distributions was checked by the Kolmogorov–Smirnov test for all tested fitness and anthropometric variables, while Levene’s test was used to test the homoscedasticity of the variables. Since all variables met the normality assumption, means and standard deviations were presented as descriptive statistics.

The effects of the applied intervention were evaluated by two-way analysis of variance for repeated measurements (ANOVA), with “Time” (pre- to post-measurement) and “Group” (Intervention vs. Control) as the main ANOVA effects and “Time × Group” as the interaction effect. To evaluate the effect size of the difference (ES), partial eta squared (µ^2^) was calculated and interpreted (small ES: >0.02; medium ES: >0.13; large ES: >0.26). Additionally, to evaluate univariate differences between measurements, a *t*-test for the dependent sample was used. Statistica 13.5 (TIBCO Software Inc., Palo Alto, CA, USA) was used, and a significance level of *p* < 0.05 was applied for all calculations.

## 3. Results

Descriptive statistics of the pre-and post-testing for the total sample (i.e., not dividing boys from girls) and the significance of the *t*-test differences between pre-and post-testing are presented in Table 1.

Table 2 shows the ANOVA results for total sample. Significant main effects for “Group” are evidenced for body height (small ES) and pacer test (small ES). Significant main effects for “Time” were evidenced for body height (large ES), body mass (small ES), broad jump (large ES), and sit-ups (medium ES). Significant interaction effect (time × group) was found for BMI (small ES) and pacer test as a measure of CRF (small ES). While observing descriptive statistics (Table 1) and ANOVA results (Table 2), it is clear that the intervention group improved their CRF, while the CRF of the control group decreased over the study period. Also, the intervention group retained their BMI levels at pre-test values, while the BMI levels of the control group slightly increased from pre- to post-testing.

The descriptive statistics of the pre- and post-testing among boys, with univariate differences between pre- and post-testing, are shown below in Table 3.

For boys, significant ANOVA main effects for “Group” were evidenced in body height (small ES), broad jump (small ES), sit-and-reach flexibility test (small ES), and sit-ups as a test of abdominal strength (small ES). Significant main effects for “Time” were found for body height (large ES), body mass (small ES), BMI (small ES), broad jump (large ES), sit-and-reach flexibility test (small ES), and sit-ups (medium ES). There were no significant Group × Time effects (Table 4). As a result, we can not specify the effects of the intervention program among boys.

Table 5 presents descriptive statistics for the pre- and post-testing in girls, and the *t*-test differences between measurements.

ANOVA main effects for “Group” in girls were significant for body height (small ES), and pacer test (small ES), while the main effects for “Time” were significant for body height (large ES), body mass (small ES), broad jump (large ES), and sit-ups (medium ES). Interaction effects (tome x group) were significant for the pacer test only (small ES) (Table 6). When comparing ANOVA results with descriptive statistics (Table 5), it is evident that the intervention group achieved improvement in CRF (pacer test), while the CRF of the control group negatively changed over the study course.

## 4. Discussion

This study assessed the impact of a 12-week online video-based PL intervention on physical fitness (PF) among high-school adolescents. The results revealed several important findings. First, applied intervention induced positive changes in CRF; this was particularly evident for girls. Second, the intervention group retained their BMI levels at the pre-testing level, while the BMI levels of the control group increased over the course of the study. Finally, no effects on observed motor capacities were evidenced. As a result, our initial study hypothesis is partially accepted.

### 4.1. Educational Intervention and Its Effects on CRF

CRF status is a highly important determinant of health status, and therefore, the improvement in the pacer test as a result of applied intervention is highly encouraging. This is particularly important knowing that this study was conducted during the period of the COVID-19 pandemic. Specifically, although in the specific study period lockdown was not declared in Croatia, sports activities were significantly reduced. This had a negative impact on the fitness status of all age groups, including adolescents, and studies repeatedly informed on negative trends in PF status of adolescents because of reduced possibilities of PA during the period of the pandemic [21,22]. Therefore, the reasons for the positive effects of applied intervention on CRF in high-school adolescents deserve attention.

The first explanation is related to the content of the educational materials applied during the intervention. Being long-time professionals in sports and physical education, the authors of the study are aware that Croatian adolescents have a low level of knowledge and understanding of certain issues related to PF, which was particularly the case for CRF. Also, the results of the current studies showed alarmingly negative trends in CRF status among Croatian adolescents [11,23]. Therefore, and as presented in the Methods section, the topic of CRF was covered with three video clips, explaining the importance of CRF, and displaying CRF training methods and modalities. The idea was to raise the awareness of the adolescents on positive effects of CRF training on overall health and well-being, including cognitive function and memory [24]. While the results of the study showed positive results, particularly regarding the improvement in CRF, it seems that applied intervention was effective.

A second explanation of the effects of the changes in CRF is associated with BMI trends (no changes in intervention group over the course of the study, and an increase in the BMI of the controls), which is explained in more detail in the following discussion. For a moment, it is important to highlight influence of BMI on endurance capacities. Specifically, BMI is negatively correlated with endurance capacities evaluated throughout field tests simply because a higher BMI implies “higher body mass”, and therefore asks for a higher oxygen consumption, which is later translated into results in field tests, such as the one we applied (e.g., pacer test) [25]. Therefore, the fact that the intervention group retained BMI values at pre-testing levels could have a certain impact on CRF results.

### 4.2. Educational Intervention and Its Effects on BMI

BMI is generally considered to be a valuable indicator of overweight/obesity status [26]. Although often being criticized in its applicability and relevancy (changes in body fat and muscle mass both influence changes in body mass, and consequently change BMI), due to its simplicity and non-invasive character of measurement, BMI is still one of the most common indicators of metabolic health [27]. Control of BMI is imperative in adolescence. Namely, in this period of life, body weight rapidly increases (particularly in girls), and therefore, there is a significant risk of overweight [28]. Therefore, the fact that BMI did not change in the intervention group, while BMI increased in the control group, should be observed as an important finding of this study. In explaining the mechanism that led to such results, two explanations emerge, both related to applied educational intervention.

As presented in the Methods section, the educational intervention consisted of various topics related to PA and PL, but also of selected themes related to diet, nutrition, and nutritional habits. Although we did not specifically evidence the eventual changes in nutritional habits of the adolescents, there is a certain possibility that participants from the intervention group accepted the information provided throughout their education and implemented it in their dietary regime. It could naturally have positive repercussions on their daily nutritional regime, and consequently, their BMI. By all means, this explanation should be additionally supported by a direct evaluation of the effects on nutrition status in further studies.

The second possible explanation for this is related to previously discussed changes in CRF status. Namely, there is a certain possibility that positive changes in CRF status were indirectly translated to the body weight management of our participants. In brief, previous studies suggested that the improvement in CRF should not be exclusively considered a “consequence of increased physical activity”, but the opposite causality is also possible. In other words, it is possible that “improved CRF could be the cause of higher physical activity”, and therefore improvement in CRF could result in an overall increase in the PA of the adolescents (i.e., they were more physically active in general) [29]. The higher PA (because of higher CRF) could be later translated into higher energy expenditure, which positively influenced the BMI (through the control of body mass) of the intervention group as well.

### 4.3. Lack of Intervention Effects on Motor Capacities

Strength, power, and flexibility capacities (motor capacities (MCs)) are important determinants of PF [30]. The quality of motor function together with body composition are directly transferred to PF and consequently to health status [31]. Moreover, MCs are an important factor in a healthy lifestyle because better MC implies higher self-confidence in physically demanding activities, and therefore, it is positively associated with a physically active lifestyle in all ages [32]. Because of the dramatic changes in life habits, the necessity of preservation and the development of MC became particularly important in the period of the COVID-19 pandemic since studies clearly noted a deterioration in horizontal jumping capacity, handgrip strength, abdominal muscle strength (sit-ups), and upper body strength (push-ups) in this period [33,34]. Therefore, one of the ideas of this investigation was that applied intervention will have certain positive effects on MC as well. However, no significant intervention effects were evidenced in that manner.

A possible explanation for the lack of intervention effects on MC could be found in the fact that we studied older adolescents. In this period of life, MC changes dramatically irrespective of training [35]. Specifically, a study on individuals aged 6–80 years from the US that aimed to develop strength growth charts noted that body-mass-normalized muscle strength evaluated by hand grip strength grew gradually from the 6th year until the 25th and 15th year of age for men and women, respectively [36]. Moreover, a similar trend was observed in abdominal muscle strength and endurance (i.e., plank test) in US youth aged 3–15 years, with the most pronounced linear growth in test results from 6 to 15 years [37]. Additionally, a large cohort study on European children and adolescents aged 9–17 years also noted a linear growth in fitness parameters, including CRF, agility, flexibility, and musculoskeletal fitness tests (sit-ups, handgrip strength tests, standing broad jump) [38]. Previously mentioned strength growth charts could lead to the conclusion that in our participants, MC developed “independent of exercise”, and that growth and development override the intervention stimulus on MC, resulting in similar changes in both study groups. Of course, a similar explanation is possible for CRF, but CRF is more (positively) influenced by changes in specific body dimensions (i.e., size of the heart, lung capacity), which occurs in an earlier life period [39].

Also, it must not be ignored that our participants regularly participated in PE lessons over the study course. Therefore, they were all involved in a similar (identical) type of physical exercising, aimed at the improvement in MC during PE classes, which consequently could contribute to similar development of MC in control and intervention groups. Logically, one can argue that such an explanation could put into question previous discussions on CRF in which intervention was effective, but from our perspective, this is not likely. In brief, CRF is developed throughout (relatively) extensive training stimuli [39]. Logically, PE classes do not provide the opportunity for such exercise types, simply because the PE curriculum in Croatia includes various themes and topics the teachers have to accomplish during lessons. On the other hand, physical exercise aimed at MC is regularly implemented in each PE lesson, mostly because of the organizational issues (non-time-consuming) and the fact that schools regularly have the necessary equipment (i.e., mats, obstacles, medicine balls). Moreover, regarding the context of the COVID-19 pandemic, the increased use of technology tools for the online education of students has been recorded, and they changed the students’ learning perceptions, which also potentially led to decreases in specific movement behaviors responsible for improving MC [40,41,42]. Specifically, a study on French children and adolescents reported a decrease in PAL together with an increase in sitting time, with a 62% increase in screen time among adolescents [43]. As a support to our general findings, we can highlight that our results are actually in agreement with a previous Canadian study that evaluated a physical-literacy-based intervention program (12-week movement skills program) on Canadian students aged 17.85 ± 0.51 years and recorded an improvement in CRF (also measured by the Pacer test), while there were no improvements in MC (standing long jump and handgrip strength) [14].

### 4.4. Limitations and Strengths

In this study, we applied field-based tests only, which is probably the most important limitation of the investigation. However, we observed relatively large groups of participants, and therefore, usage of laboratory-based testing was limited. Also, other than PF and anthropometrics, we did not gather information on other important health determinants (i.e., PAL, metabolic health, nutritional intake). Next, we cannot ignore the fact that some intervention effects were minimal in magnitude, but still very promising. It is important to note that participants were not (personally) randomly assigned to study groups, but division in groups was carried out by random selection of school classes. Finally, the study was conducted in relatively uncontrolled settings, and because of the study design and type of intervention, we were not able to absolutely equalize the conditions for control and intervention.

This is one of the rare studies in which the effects of the education interventions oriented toward PL were studied in the context of its efficacy in PF. A gender-stratified approach, informed by our findings of gender differences, stands as a significant strength of this investigation. Finally, to the best of our knowledge, this is the first study in which the effects of such an intervention were evidenced in the period of the COVID-19 pandemic. Knowing the detrimental effects of the pandemic period on the fitness and health indices of adolescents, we hope that the study will initiate further analyses.

## 5. Conclusions

The recent COVID-19 pandemic highlighted the necessity of alternative approaches when applying interventions aimed at the improvement in PF status. Consequently, various forms of online-based education in different fields have become regular and convenient methods of education. In this study, we applied specific online video-based education aimed at an improvement in PF among high-school adolescents with encouraging results. It was hypothesized and expected that the intervention group would improve their PF as a consequence of the applied PL education, which was partially confirmed.

The positive effects of the applied 12-week educational intervention on BMI and CRF are highly important knowing that both indices are related not only to PF, but also to health status. However, it seems that the applied intervention was more effective for girls than for boys. This is one of the rare studies in which the effects of PL education were studied on its efficacy in improving PF. However, this study did not include some important determinants of health status, such as physical activity habits, nutrition and consumption of psychoactive substances (e.g., alcohol and tobacco), which should be incorporated in future similar studies.

## Figures and Tables

**Figure 1 children-10-01666-f001:**
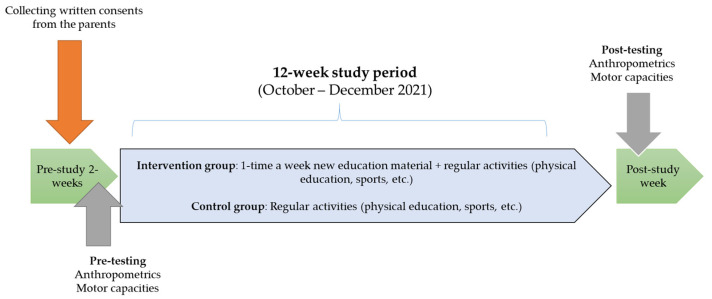
Study design with timeframes.

**Table 1 children-10-01666-t001:** Descriptive statistics for pre- and post-testing for the total sample with data given as means ± standard deviations and univariate differences between pre- and post-testing results (* indicates the statistical significance of the differences between measurements).

	Total Sample (*n* = 423)		Control (*n* = 193)		Intervention (*n* = 230)	
	Pre-Test	Post-Test	Pre-Test	Post-Test	Pre-Test	Post-Test
Body height (cm)	171.21 ± 8.86	172.98 ± 9.09 *	170.05 ± 8.39	170.48 ± 8.58 *	172.32 ± 9.17	173.23 ± 9.33 *
Body mass (kg)	64.08 ± 14.71	64.95 ± 14.35 *	64.38 ± 16.02	65.06 ± 15.73	63.79 ± 13.36	63.82 ± 13.11 *
BMI (kg/m^2^)	21.73 ± 4.11	21.69 ± 3.76	21.67 ± 4.74	22.27 ± 4.18 *	21.36 ± 3.36	21.36 ± 3.37
Broad jump (cm)	182.03 ± 32.91	187.18 ± 32.38 *	183.07 ± 34.26	190.88 ± 32.63 *	181.04 ± 31.59	184.33 ± 31.97 *
Sit and Reach (cm)	11.25 ± 10.91	11.69 ± 7.47	11.04 ± 7.66	12.14 ± 6.76 *	11.46 ± 13.33	11.83 ± 8.01 *
Sit-ups (rep)	56.41 ± 14.46	62.05 ± 15.52 *	56.58 ± 16.02	63.38 ± 17.98 *	56.24 ± 12.81	61.02 ± 13.29 *
Pacer test (level)	9.06 ± 3.17	9.14 ± 3.33	9.21 ± 3.39	9.13 ± 3.85	9.34 ± 2.93	9.99 ± 2.76 *

**Table 2 children-10-01666-t002:** Results of the two-way analysis for repeated measurement for total sample (µ^2^—partial eta squared values—effect size).

	Main Effects						Interaction (Group × Time)		
	Group			Time					
	F Test	*p*	η^2^	F Test	*p*	η^2^	F Test	*p*	η^2^
Body height	9.6	0.01	0.02	162	0.001	0.28	1.2	0.28	0.002
Body mass	0.25	0.62	0.001	14	0.001	0.03	2.43	0.12	0.01
BMI	0.99	0.32	0.002	3.32	0.07	0.008	3.94	0.04	0.02
Broad jump	2.25	0.13	0.01	7117.5	0.001	0.95	0.57	0.45	0.002
Sit and Reach	0.06	0.81	0.001	0.27	0.6	0.001	1.84	0.18	0.004
Sit-ups	2.78	0.1	0.01	76.75	0.001	0.17	0.05	0.83	0.001
Pacer test	5.71	0.02	0.02	0.86	0.36	0.002	23.71	0.001	0.06

**Table 3 children-10-01666-t003:** Descriptive statistics for pre- and post-testing for boys with data given as means ± standard deviations, and univariate differences between pre- and post-testing results (* indicates statistical significance of the differences between measurements).

Boys	Total Sample (*n* = 128)		Control (*n* = 53)		Intervention (*n* = 75)	
	Pre-Test	Post-Test	Pre-Test	Post-Test	PRE-Test	Post-Test
Body height (cm)	180.02 ± 7.41	181.33 ± 7.07 *	178.46 ± 7.34	179.74 ± 6.79 *	181.3 ± 7.26	182.45 ± 7.09 *
Body mass (kg)	72.9 ± 15.91	73.31 ± 14.44 *	72.74 ± 17.68	76.4 ± 16.54 *	73.03 ± 14.38	73.68 ± 12.75 *
BMI (kg/m^2^)	22.44 ± 4.45	22.72 ± 4.03 *	22.79 ± 5.11	23.58 ± 4.57 *	22.15 ± 3.83	22.12 ± 3.5
Broad jump (cm)	211.89 ± 29.75	215.68 ± 27.85 *	216.85 ± 29.52	222.78 ± 24.25 *	207.79 ± 29.48	210.77 ± 29.25 *
Sit and Reach (cm)	7.36 ± 8.07	8.78 ± 8.57 *	8.52 ± 7.31	10.96 ± 7.5 *	6.4 ± 8.57	7.27 ± 8.98 *
Sit-ups (rep)	66.62 ± 13.83	71.48 ± 15.25 *	68.9 ± 15.99	75.04 ± 18.64 *	64.74 ± 11.51	69.03 ± 11.93 *
Pacer test (level)	11.62 ± 3.23	11.94 ± 3.26	11.8 ± 3.74	11.87 ± 4.01	11.47 ± 2.75	11.98 ± 2.64

**Table 4 children-10-01666-t004:** Results of the two-way analysis for repeated measurement for boys (µ^2^—partial eta squared values—effect size).

	Main Effects						Interaction (Group × Time)		
	Group			Time					
	F Test	*p*	η^2^	F Test	*p*	η^2^	F Test	*p*	µ η^2^
Body height	4.41	0.04	0.03	66.00	0.001	0.34	0.31	0.58	0.002
Body mass	0.33	0.56	0.003	11.00	0.001	0.08	3.4	0.07	0.03
BMI	2.26	0.14	0.02	4.35	0.04	0.03	3.55	0.06	0.03
Broad jump	6.61	0.01	0.05	39.5	0.001	0.37	1.7	0.19	0.01
Sit and Reach	5.31	0.02	0.04	12.2	0.001	0.09	0.89	0.35	0.01
Sit-ups	6.24	0.01	0.05	25.42	0.001	0.18	0.01	0.91	0.001
Pacer test	0.17	0.69	0.001	0.12	0.73	0.001	2.69	0.1	0.02

**Table 5 children-10-01666-t005:** Descriptive statistics for pre- and post-testing for girls with data given as means ± standard deviations, and univariate differences between pre- and post-testing results (* indicates statistical significance of the differences between measurements).

Girls	Total Sample (*n* = 295)		Control (*n* = 140)		Intervention (*n* = 195)	
	Pre-Test	Post-Test	Pre-Test	Post-Test	Pre-Test	Post-Test
Body height (cm)	167.43 ± 6.41	167.92 ± 6.51 *	166.84 ± 6.28	166.98 ± 6.27 *	168.03 ± 6.5	168.77 ± 6.62 *
Body mass (kg)	60.3 ± 12.4	59.67 ± 11.66 *	59.37 ± 10.26	59.93 ± 10.74	61.19 ± 14.13	60.38 ± 12.63
BMI (kg/m^2^)	21.42 ± 3.92	21.1 ± 3.54	21.84 ± 4.57	21.22 ± 3.84	20.98 ± 3.06	21 ± 3.25
Broad jump (cm)	169.19 ± 24.92	173.28 ± 24.36 *	170.11 ± 26.19	176.67 ± 24.97 *	168.26 ± 23.59	170.53 ± 23.58 *
Sit and Reach (cm)	12.92 ± 11.54	12.96 ± 6.56	12 ± 7.6	12.58 ± 6.43	13.88 ± 14.49	13.49 ± 6.69
Sit-ups (rep)	52.02 ± 12.38	57.44 ± 13.45 *	51.88 ± 13.33	58.19 ± 15.07 *	52.17 ± 11.35	56.84 ± 12.02 *
Pacer test (level)	7.96 ± 2.43	8.23 ± 2.62 *	8.01 ± 2.4	7.71 ± 2.83	8.33 ± 2.42	8.95 ± 2.2 *

**Table 6 children-10-01666-t006:** Results of the two-way analysis for repeated measurement for girls (µ^2^—partial eta squared values—effect size).

	Main Effects						Interaction (Group × Time)		
	Group			Time					
	F Test	*p*	η^2^	F Test	*p*	η^2^	F Test	*p*	η^2^
Body height	5.5	0.02	0.02	108	0.001	0.27	0.4	0.52	0.001
Body mass	0.18	0.67	0.001	6.3	0.01	0.02	0.1	0.78	0.001
BMI	0.17	0.68	0.001	0.14	0.71	0.001	1.04	0.31	0.004
Broad jump	1.04	0.31	0.004	33.6	0.001	0.36	0.31	0.58	0.001
Sit and Reach	2.15	0.14	0.01	0.17	0.68	0.001	1.56	0.21	0.01
Sit-ups	0.9	0.34	0.004	20.81	0.001	0.17	0.03	0.86	0.001
Pacer test	13.2	0.001	0.05	0.75	0.39	0.003	15.03	0.01	0.09

## Data Availability

Data are available upon reasonable request.

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
