# Peer review of "Effectiveness of Physical-Literacy-Based Online Education on Indices of Physical Fitness in High-School Adolescents: Intervention Study during the COVID-19 Pandemic Period"

_children, 2023, doi:10.3390/children10101666_

Round 1

Reviewer 1 Report

Dear Authors,

Thank you very much for your interesting work. After a thorough review, I do believe that your paper requires only minor revisions. Specific comments and suggestions for improvement can be found below. I believe these changes will further enhance the clarity and impact of your paper.

This study aimed to assess the impact of a specific online video-based physical literacy (PL) intervention on physical fitness (PF) among high school students in Croatia. Involving 423 adolescents, the 12-week intervention utilized educational video materials disseminated through a closed social network. The results demonstrated significant positive effects on cardiorespiratory fitness (CRF) and body mass index (BMI), with particularly notable improvements in girls. These findings highlight the potential of educational interventions to positively influence health-related PF indicators, which is of considerable importance for adolescent well-being.

This paper appears to align well with the scope of the journal, which focuses on various aspects of physical fitness, health, and related interventions. The study explores the impact of a physical literacy intervention on physical fitness in high school students, directly relating to the journal's area of interest in promoting health and fitness among different populations.

I recommend incorporating discussions about the changes that occurred during the COVID-19 pandemic, particularly the increased use of technological tools by students in online education. This could be supported by citing a relevant reference (e.g., 10.3390/educsci12080573).

Furthermore, the pandemic period has spurred the development of new teaching methods, and it's essential to acknowledge that student perceptions play a crucial role in this transformation, as highlighted in a recent study (Reference: 10.1016/j.jevs.2023.104537).

Specific comments:

Line 100-101: For clarity, consider: "In this intervention study, 423 high-school students from Osijek-Baranja County in northern Croatia participated."

Line 103-104: Slightly adjust for flow: "The students were divided into an intervention group (n = 230) and a control group (n = 193), matched by gender using random class sampling."

Line 113-114: To better adhere to formal tone: "The study was approved by the Ethical Board of the Kinesiology University of Zagreb, Croatia (Ref. no.25/2021, date of approval: 16 July 2021)."

Line 116: For completeness: "The study design, including group assignments and interventions, is detailed in Figure 1."

Line 163-164: Consider consolidating to: "This study's variables, aside from the study group (intervention vs. control) and gender (male vs. female), were anthropometrics and indices of PF."

Line 263-264: Reword to "This study assessed the impact of a 12-week online video-based PL intervention on physical fitness (PF) among high-school adolescents."

Line 266: Suggest changing "which was" to "this was".

Line 346-353: Consider rephrasing for clarity and brevity, discussing the age-related developments and referencing the relevant studies without delving too deep into other age groups

Line 392-393: Highlight the novel approach: "A gender-stratified approach, informed by our findings of gender differences, stands as a significant strength of this investigation."

Author Response

Dear Authors,

Thank you very much for your interesting work. After a thorough review, I do believe that your paper requires only minor revisions. Specific comments and suggestions for improvement can be found below. I believe these changes will further enhance the clarity and impact of your paper.

This study aimed to assess the impact of a specific online video-based physical literacy (PL) intervention on physical fitness (PF) among high school students in Croatia. Involving 423 adolescents, the 12-week intervention utilized educational video materials disseminated through a closed social network. The results demonstrated significant positive effects on cardiorespiratory fitness (CRF) and body mass index (BMI), with particularly notable improvements in girls. These findings highlight the potential of educational interventions to positively influence health-related PF indicators, which is of considerable importance for adolescent well-being.

This paper appears to align well with the scope of the journal, which focuses on various aspects of physical fitness, health, and related interventions. The study explores the impact of a physical literacy intervention on physical fitness in high school students, directly relating to the journal's area of interest in promoting health and fitness among different populations.

RESPONSE: Thank you very much for your support and recognizing the importance and quality of our work! We tried to amend the manuscript according to your suggestions.

I recommend incorporating discussions about the changes that occurred during the COVID-19 pandemic, particularly the increased use of technological tools by students in online education. This could be supported by citing a relevant reference (e.g., 10.3390/educsci12080573).

Furthermore, the pandemic period has spurred the development of new teaching methods, and it's essential to acknowledge that student perceptions play a crucial role in this transformation, as highlighted in a recent study (Reference: 10.1016/j.jevs.2023.104537).

RESPONSE: Thank you for this suggestion. It is now included in the discussion. Text reads: “). Moreover, regarding the context of the COVID-19 pandemic, the increased use of technology tools for online education of students has been recorded and they changed the students learning perceptions, which also potentially led to decreases in specific movement behaviors responsible for improving MC [49-51]. Specifically, a study on French children and adolescents reported decrease in PAL together with increase in sitting time, with 62% increase in screen time among adolescents [52]”. Please see 4.3. Lack of intervention effects on motor capacities paragraph in the Discussion.

Specific comments:

Line 100-101: For clarity, consider: "In this intervention study, 423 high-school students from Osijek-Baranja County in northern Croatia participated."

RESPONSE: Thank you for this suggestion, amended accordingly.

Line 103-104: Slightly adjust for flow: "The students were divided into an intervention group (n = 230) and a control group (n = 193), matched by gender using random class sampling."

RESPONSE: Amended accordingly.

Line 113-114: To better adhere to formal tone: "The study was approved by the Ethical Board of the Kinesiology University of Zagreb, Croatia (Ref. no.25/2021, date of approval: 16 July 2021)."

RESPONSE: Amended accordingly.

Line 116: For completeness: "The study design, including group assignments and interventions, is detailed in Figure 1."

RESPONSE: Amended accordingly.

Line 163-164: Consider consolidating to: "This study's variables, aside from the study group (intervention vs. control) and gender (male vs. female), were anthropometrics and indices of PF."

RESPONSE: Amended accordingly.

Line 263-264: Reword to "This study assessed the impact of a 12-week online video-based PL intervention on physical fitness (PF) among high-school adolescents."

RESPONSE: Amended accordingly.

Line 266: Suggest changing "which was" to "this was".

RESPONSE: Amended accordingly.

Line 346-353: Consider rephrasing for clarity and brevity, discussing the age-related developments and referencing the relevant studies without delving too deep into other age groups

RESPONSE: Thank you for this suggestion but the aim of this part of the text was to emphasize that MC changes irrespective of training. Thus, we intentionally included detailed information about age groups that support our hypothesis, and we believe this is an important part of this section of Discussion.

Line 392-393: Highlight the novel approach: "A gender-stratified approach, informed by our findings of gender differences, stands as a significant strength of this investigation."

RESPONSE: It is now included in the Limitations and Strengths section.

Thank you again for your comments and support!

Reviewer 2 Report

In this study, a substantial cohort of 423 adolescents was enrolled to investigate the impact of physical activity within an intervention group on various measures of 'fitness' when compared to a control group.

In a comprehensive manuscript, the authors conclude that the intervention yields benefit, particularly within the subgroup of girls. However, it's noteworthy that within this subgroup, the observed improve­ment is confined solely to cardiorespiratory fitness, with a relatively small effect size (µ² = 0.02).

As a reviewer, I hold the conviction that sedentary behaviour is strongly linked to the development of numerous diseases. Therefore, it is imperative to take all possible measures to reduce such behaviour. In this context, it is essential to note that the study does not provide information on whether the adolescents included in the study were exposed to extended periods of sedentary activities, such as increased homework demands or cancelled school hours, which could potentially contribute to the observed changes.

Below, you will find some major and minor comments for your consideration.

Major comments

Introduction: Somewhat lengthy. Could be shortened.

Intervention program: who has the developer been? Any compony; any authority?

l. 95: Maybe delete ‘Initially’ because it suggests that after having seen the results, you developed a secondary hypothesis.

l. 162: Were have the measurements been performed?

l. 162: No dropouts over 12 weeks with so many participants?

l. 165: Human length shrinks during a day by about 3 cm. Thus, provide constant time of measurement.

l. 166 -177: don’t exaggerate. Readers know how height and mass are assessed and how the BMI is calculated. Please, shorten. Furthermore: In Table 1 you present two decimals but you have measured solely one decimal. Please, correct.

Λ. 252: Table 5. BMI after intervention with no asterisk?

l. 266: The difference in the BMI is .60 BMI units. Please, comment the significance. In particular, as in the entire group, also controls improved over time, except for the pacer test.

l. 410: … so many other studies are needed in this world, i.e. this sentence does not make much sense.

Minor comments.

Fig. 1: Please, explain PE in the legend.

l. 106-108: these are results.

l. 222: All and Group is identical? If yes, unify. Relates also to ‘total sample’.

Table 3: Please, not again All. Maybe All Boys is more precise.

l. 231: this is eta η, and this is mu: µ

Use of the English language is fine. Only punctuation here and there would improve the readability.

Author Response

In this study, a substantial cohort of 423 adolescents was enrolled to investigate the impact of physical activity within an intervention group on various measures of 'fitness' when compared to a control group.

RESPONSE: Thank you for your support and for recognizing the importance of our work! We tried to amend the manuscript according to your suggestions.

In a comprehensive manuscript, the authors conclude that the intervention yields benefit, particularly within the subgroup of girls. However, it's noteworthy that within this subgroup, the observed improvement is confined solely to cardiorespiratory fitness, with a relatively small effect size (µ² = 0.02).

RESPONSE: Thank you for this suggestion. Indeed, the changes were minimal, and ES you mentioned was relatively small. However, the ES of interaction effect includes changes and differences in pre- to post-changes of both groups. Therefore, we believe this is worthy finding, especially if we consider the fact that intervention was done “specifically” throughout “theoretical learning”, while we directly observed changes in exact measures of PF. However, your observation is now included in the Limitations. The text reads: “Next, we can’t ignore the fact that some intervention effects were minimal in magnitude, but still very promising” (please see Limitations and Strengths subsection).

As a reviewer, I hold the conviction that sedentary behaviour is strongly linked to the development of numerous diseases. Therefore, it is imperative to take all possible measures to reduce such behaviour. In this context, it is essential to note that the study does not provide information on whether the adolescents included in the study were exposed to extended periods of sedentary activities, such as increased homework demands or cancelled school hours, which could potentially contribute to the observed changes.

RESPONSE: Thank you for this observation. Indeed, we did not specifically assess sedentary behaviours as physical activity was the focus of this research and we believe it can display at least that part of movement behaviours. We will definitely strive to include sedentarism variables in future research, thank you. However, the explanation about possible alterations in sedentary behaviours is added in the Discussion, text reads: “Moreover, regarding the context of the COVID-19 pandemic, the increased use of technology tools for online education of students has been recorded and they changed the students learning perceptions, which also potentially led to decreases in specific movement behaviors responsible for improving MC [49-51]. Specifically, a study on French children and adolescents reported decrease in PAL together with increase in sitting time, with 62% increase in screen time among adolescents [52]”.

Below, you will find some major and minor comments for your consideration.

Major comments

Introduction: Somewhat lengthy. Could be shortened.

RESPONSE: It is now shortened, please see changes throughout Introduction.

Intervention program: who has the developer been? Any compony; any authority?

RESPONSE: The “developer” was a team of experts in the field. It is now explained in the Methods section, text reads: “The developers of the intervention program were experts in each field and topic of the video material (i.e., strength and conditioning coach, medical doctor, nutritionist and experienced PE teacher).”

  1. 95: Maybe delete ‘Initially’ because it suggests that after having seen the results, you developed a secondary hypothesis.

RESPONSE: Amended accordingly.

  1. 162: Were have the measurements been performed?

RESPONSE: Thank you for noticing this. Text reads: “All measurements and assessments have been conducted in the school facilities (i.e., PE gym).”

  1. 162: No dropouts over 12 weeks with so many participants?

RESPONSE: Thank you for noticing this reporting error! Initially, we included 523 participants, and the final/total number was 423 who participated in both initial and final testing. It is now included in the participants and study design section. Text reads: “In this intervention study, 423 high-school adolescents (295 girls, 128 boys; 14-to-18 years of age) from Osijek-Baranja County in northern Croatia participated in both initial and final testing points. At the initial point of the assessment, 523 participants (366 girls, 153 boys153 males, and 366 females) were included”.

  1. 165: Human length shrinks during a day by about 3 cm. Thus, provide constant time of measurement.

RESPONSE: Thank you for noticing that this is missing, it is now added in the text: “All measurements and assessments have been conducted in the school facilities (i.e., PE gym) during the morning (8:00-10:00 AM) to avoid diurnal variations.”

  1. 166 -177: don’t exaggerate. Readers know how height and mass are assessed and how the BMI is calculated. Please, shorten. Furthermore: In Table 1 you present two decimals but you have measured solely one decimal. Please, correct.

RESPONSE: It is now removed and corrected, thank you.

Λ. 252: Table 5. BMI after intervention with no asterisk?

RESPONSE: Yes, we evidenced no effects for BMI in girls. Almost certainly, the number of subjects reduced the possibility to reach the statistical significance despite some numerical differences between pre- and post-measurement

  1. 266: The difference in the BMI is .60 BMI units. Please, comment the significance. In particular, as in the entire group, also controls improved over time, except for the pacer test.

RESPONSE: Indeed, the difference between pre- and post-measurement for total sample was significant, but when observed for boys and girls no significant Time effects were found. Therefore, we would rather not strongly elaborate about “significant effects” on BMI, especially if we consider the fact that BMI is influenced both by lean body mass, and body fat. However, if you would insist on profound discussion we will certainly follow your advice. Thank you!

  1. 410: … so many other studies are needed in this world, i.e. this sentence does not make much sense.

RESPONSE: Thank you, it is now removed.

Minor comments.

Fig. 1: Please, explain PE in the legend.

RESPONSE: Amended accordingly, it is now changed to physical education within the figure.

  1. 106-108: these are results.

RESPONSE: Yes, maybe it seems like that, but we only wanted to note no difference between groups in out-of-school sport participation, for better results interpretation. We will remove the numbers to avoid misunderstanding.

  1. 222: All and Group is identical? If yes, unify. Relates also to ‘total sample’.

RESPONSE: Group relates to Control and Intervention. We changed All into Total sample to clarify this issue.

Table 3: Please, not again All. Maybe All Boys is more precise.

RESPONSE: It is changed to Total sample, and we indicated whether it relates to total sample of boys or girls.

  1. 231: this is eta η, and this is mu: µ

RESPONSE: Changed throughout the tables, thank you for noticing.

Staying at your disposal!

Reviewer 3 Report

The title is clear and describes the content of the manuscript.

The summary also complies with the corresponding sections.

The introduction and justification clearly state the background of the problem, as well as those issues that are unclear or require resolution.

Regarding the objectives, make a better definition of them to clarify the purpose of the study, as well as its contribution. Therefore, it is recommended to expand the main objective, as well as reformulate the secondary objectives so that they describe in a more detailed way the variables that are going to be analyzed.

It is recommended to expand the conclusions section and include the expected and achieved results of the program, as well as its strengths and weaknesses.

The main conclusions are repeated at the beginning and at the end of the Discussion section. It is recommended to put them at the end to avoid redundant information in the same section.

It is recommended not to include subsections in the Discussion. It is preferable that the main study findings connected to the objectives be presented.

Author Response

The title is clear and describes the content of the manuscript.

RESPOND: Thank you.

The summary also complies with the corresponding sections.

RESPOND: Thank you.

The introduction and justification clearly state the background of the problem, as well as those issues that are unclear or require resolution.

RESPOND: Thank you.

Regarding the objectives, make a better definition of them to clarify the purpose of the study, as well as its contribution. Therefore, it is recommended to expand the main objective, as well as reformulate the secondary objectives so that they describe in a more detailed way the variables that are going to be analyzed.

RESPOND: It is now reformulated, and specific details have been added. Text reads:  “The aim of this study was to evaluate the effectiveness of specific PL-intervention (e.g., video-based-education) on PF indices (CRF, flexibility, muscular strength and power) in high school students during the  pandemic period. Knowing the specifics and differences of PF between genders, in this study we applied a gender-stratified approach in studying the effects of PL-intervention on PF in high-school adolescents. We hypothesized that applied intervention would have positive effects on PF irrespective of gender. This research would give insight into how PL intervention impacts PF including CRF, flexibility, muscular strength and power, which are important determinants of health status in general.”

It is recommended to expand the conclusions section and include the expected and achieved results of the program, as well as its strengths and weaknesses.

RESPOND: The conclusion is now expanded as you specified. Please see the entire Conclusion section; thank you.

. It is recommended to put them at the end to avoid redundant information in the same section.

RESPOND: We tried to reformulate parts of the discussion and cocnlusion accordingly (please see previous comment also). Thank you for your suggestion. 

It is recommended not to include subsections in the Discussion. It is preferable that the main study findings connected to the objectives be presented.

RESPOND: We believe that it is much easier to follow the discussion if it has certain subheadings, that is why we divided the text like this. But, if you will insist on removing it, we will certainly follow the requirement. Staying at your disposal.

Staying at your disposal!

Round 2

Reviewer 1 Report

Dear Authors,

I wanted to extend my heartfelt congratulations to you and your team for the outstanding job you've done in revising your paper. I am genuinely impressed by the way you have meticulously incorporated the suggested revisions. Your commitment to improving the article's quality is evident, and I must say that the final result is nothing short of exceptional. The transformation from the initial draft to the current version is remarkable and a testament to your dedication to excellence.

Reviewer 2 Report

Thanks for the comprehensively revised version.